# Silk/Natural Rubber (NR) and 3,4-Dihydroxyphenylalanine (DOPA)-Modified Silk/NR Composites: Synthesis, Secondary Structure, and Mechanical Properties

**DOI:** 10.3390/molecules25010235

**Published:** 2020-01-06

**Authors:** Hiromitsu Sogawa, Treratanakulwongs Korawit, Hiroyasu Masunaga, Keiji Numata

**Affiliations:** 1Biomacromolecules Research Team, RIKEN Center for Sustainable Resource Science, 2-1, Saitama, Wako 351-0198, Japan; hiromitsu.sogawa@riken.jp (H.S.); godofjj@gmail.com (T.K.); 2Materials Structure Group I, Research & Utilization Division, Japan Synchrotron Radiation Research Institute, 1-1-1, Kouto, Sayo-cho, Sayo-gun, Hyogo 679-5198, Japan; masunaga@spring8.or.jp

**Keywords:** silk, natural rubber (NR), composite, enzymatic modification, 3,4-dihydroxyphenylalanine (DOPA)

## Abstract

Silk composites with natural rubber (NR) were prepared by mixing degummed silk and NR latex solutions. A significant enhancement of the mechanical properties was confirmed for silk/NR composites compared to a NR-only product, indicating that silk can be applied as an effective reinforcement for rubber materials. Attenuated total reflection Fourier transform infrared (ATR-FTIR) and wide-angle X-ray diffraction (WAXD) analysis revealed that a β-sheet structure was formed in the NR matrix by increasing the silk content above 20 wt%. Then, 3,4-dihydroxyphenylalanine (DOPA)-modified silk was also blended with NR to give a DOPA-silk/NR composite, which showed superior mechanical properties to those of the unmodified silk-based composite. Not only the chemical structure but also the dominant secondary structure of silk in the composite was changed after DOPA modification. It was concluded that both the efficient adhesion property of DOPA residue and the secondary structure change improved the compatibility of silk and NR, resulting in the enhanced mechanical properties of the formed composite. The knowledge obtained herein should contribute to the development of the fabrication of novel silk-based elastic materials.

## 1. Introduction

Silk fibroins are structural proteins that exhibit remarkable mechanical properties, biodegradability, and biocompatibility [1,2]. Among them, silkworm silk from *Bombyx mori (B. mori)* has high processability and can be processed into various forms, including fibers, nano/microparticles, films, sponges, and hydrogels [3,4,5,6,7,8]. The excellent mechanical properties of silk fibroins are attributed to their unique secondary structure, that is, the coexistence of β-sheet crystal domains and an amorphous matrix. In the case of *B. mori* silk, repetitive GAGAGX hexapeptide units (X = S, Y, or A) are frequently observed in the sequence and are essential for the formation of the antiparallel β-sheet structure [9,10,11]. Owing to such a characteristic feature, useful silk-based materials have also been produced by combining silk with other materials, such as polymers [12,13,14,15,16,17,18,19,20] and inorganics [21,22,23]. Recently, we achieved tensile reinforcement and improved the water-resistant properties of silk by adding telechelic-type polyalanine [15] and fluoropolymers [14], respectively. Silk/epoxy resin composites were also prepared in our group, and their toughening mechanism was studied in detail [12,13]. For silk-based composites, the interfacial properties between silk and other materials are critical for defining their mechanical properties. The modification of silk is a facile approach to improve the interfacial adhesion properties. For instance, the tuning of silk hydrophilic/hydrophobic features was achieved via the introduction of polymer chains onto the side chains of silks [24,25,26,27,28,29], whereas the decoration of silk with catechol groups was demonstrated by mimicking the adhesion proteins that exhibit strong adhesive features toward several materials owing to the presence of this functional group in 3,4-dihydroxyphenylalanine (DOPA) residue [30,31,32]. More specifically, DOPA-modified silkworm silk from *B. mori* was reported to exhibit five-times higher adhesion strength for several surfaces, such as mica, polymer, and wood compared, to that of unmodified silk [30].

In addition, natural rubber (NR) is an indispensable natural resource for various rubber products, including latex gloves, textiles, and tires for vehicles and aircraft [33,34,35]. Despite tremendous advances in synthetic chemistry, approximately 40% of rubber products are still prepared from NR [36]. The NR latex from the *Hevea brasiliensis* tree is the main industrial resource and is known to contain *cis*-1,4-poly(isoprene), mainly in addition to a small portion of proteins, carbohydrates, lipids, and salts [37]. In most applications, NR is vulcanized and is compounded with fillers to enhance the mechanical properties. Carbon black (CB) and carbon nanotubes (CNTs) have been most widely used as useful fillers for NR since the early days of the field, and NR composites with these fillers have been continuously developed even today [38,39,40,41]. Recently, nanocelluloses, including cellulose nanofibers (CNFs) and cellulose nanocrystals (CNCs), have also been applied as bio-based reinforcements of NR in terms of the reduction of health hazards and environmental loads [42,43,44,45,46]. Several attempts have been made to incorporate nanocelluloses as reinforcing fillers in nonvulcanized NR; however, the low compatibility of NR and nanocelluloses sometimes hinders the satisfactory enhancement of the properties [47]. To improve the compatibility between nanocelluloses and rubber products, the modification of CNC surfaces with a hydrocarbon chain was demonstrated by Berglund and coworkers [48]. They prepared composites from polybutadiene (PBD) and modified CNCs and elucidated that the mechanical properties were improved owing to the intercalated structures. Despite their excellent mechanical properties, high-performance composites with NR and nanocellulose are still limited, probably because the surface modification of nanocelluloses remains a challenging research target [49].

Based on this scientific background, we expected that silk could be a new candidate for the ecofriendly reinforcement of NR and other rubber products because of its characteristics, such as high processability and ease of modification, as introduced above. A few studies on silk and NR composites have been reported to date, but the mechanical properties of formed composites have not been adequately discussed. For instance, Wang et al. described the composite preparation of NR and *B. mori* silk [50]. As their main focus was antibacterial properties of formed composites, they did not describe the detail of the mechanical properties. Rajkumar et al. studied the reinforcement of *cis*-1,4-poly(isoprene) by adding silk textile [51]. In their study, the hybrid fibers of silk and wool were applied to prepare the composites, and thus the appropriate evaluation of the reinforcement effects that are attributed to individual components should be difficult. Besides, the hybrid fibers were heterogeneously mixed with *cis*-1,4-poly(isoprene) and processed under the harsh condition, which made it difficult to discuss the interfacial interactions of silk and other materials in connection with their chemical structures. Meanwhile, modified silks, such as DOPA-modified silk, have never been applied to composites, despite their high potential to enhance the interfacial adhesion properties between DOPA structures with NR domains. The fabrication of silk/NR composites with superior mechanical properties would to be achieved by utilizing such a modification approach. In short, further study is still required for the development of a well-established silk/NR composite system. Herein, we newly prepared silk/NR composites by mixing degummed silk solution from *B. mori* cocoons and NR latex solution and elucidated their mechanical properties and secondary structures, using several instruments. Furthermore, DOPA-modified silk was also applied to a composite system with NR and the effect of DOPA modification was studied.

## 2. Results and Discussion

### 2.1. The Preparation and Basic Properties of Silk/NR Composites

The composites of silk and NR were prepared by varying the silk contents (Figure 1). After conventional mixing and drying, silk/NR composites with up to 40 wt% silk were successfully prepared. The thickness of the formed composites was approximately 1.0 mm. The transparency of the composites slightly decreased with increasing silk content, and an opaque sample, which was brittle and hard to detach from the mold, was obtained at a silk content of 60 wt%. The good mixing of silk and NR was inhibited probably because the aggregation of silk was more enhanced at this high silk content. Then, the mechanical properties of the silk/NR composites below a silk content of 40 wt% were evaluated by tensile tests with an elongation rate of 10 mm/min (Figure 2). The dumbbell-shaped samples were prepared in accordance with JIS K 6251 and subjected to tensile tests [52]. Note that all samples in this study were not vulcanized by vulcanizing agents such as sulfur; thus, there should be no chemical crosslinking points in the samples. Figure 3 summarizes the Young’s modulus, elongation at break, breaking strength, and fracture energy of the formed composites. Obviously, Young’s modulus increased with increasing silk content, whereas the elongation at break decreased. The breaking strength and fracture energy reached maxima with 20 wt% silk content, characteristic of tough silk, and elastic NR appeared most efficiently at this condition. The fracture energy, Young’s modulus, and elongation of silk_20_/NR_80_ were 35 MJ/m^3^, 38 MPa, and 750%, respectively. Siqueria et al. reported that no vulcanized NR/nanocellulose composite with 6% cellulose content showed a Young’s modulus of 100 MPa, although its elongation was only 5.9% [45]. Meanwhile, NR composites with 20 wt% CB had Young’s modulus and elongation of 47 MPa and 42%, respectively [53]. Compared with those materials, the silk/NR composites were found to exhibit a moderate modulus with good elasticity. Additionally, compared to NR/nanocellulose composites, there was a merit that a larger amount of silk could be used without decreasing the mechanical properties, probably owing to the better compatibility of silk with NR than that of nanocelluloses. Only less than 10 wt% celluloses were applied for most of the NR/nanocellulose composites to maintain the good dispersion of celluloses in the NR matrix. Thus, facile tuning of the mechanical properties in a broader range could be achieved in the silk/NR system.

The rheological properties of silk/NR composites were characterized in terms of the storage elastic modulus (*G′*) and the loss elastic modulus (*G’’*). Figure 4 shows the strain dependency of samples between 0.01% and 10% at a constant frequency (10 Hz). Note that *G’’* was smaller than *G’* for all measurements and was omitted from Figure 4 for clarity. Similar to the results of tensile tests, the significant increase in *G’* with increasing silk contents was confirmed when we focused on the *G’* values of the linear viscoelastic region. The frequency sweep measurements also gave similar results (Appendix A). Meanwhile, above 0.1% strain, a gradual decrease in *G’* values was observed for the silk/NR composites. The slope was steeper, and the decrease started at lower strain for composites with a high silk content. These changes were attributed to the introduction of a rigid silk component to the NR matrix.

The surface morphologies of the composites were imaged by scanning electron microscopy (SEM) (Figure 5). The surface morphologies of the samples were observed before stretching and after tensile deformation. Before stretching, although the roughness of the composites gradually increased with increasing silk content, large phase-separated domains were not clearly observed. This result suggests that silk was well dispersed in the NR matrix because of its good compatibility. On the other hand, the observation of fracture surfaces indicated no specific alignment of cracking and voids. Simply, the surface roughness tended to increase after tensile deformation except for the pure NR sample. The secondary structure of silk in the composites was changed by varying the silk content as described later; however, the morphological differences that reflect such changes were not imaged.

### 2.2. Structural Characterization of Silk/NR Composites

To elucidate the secondary structures of silk/NR composites, attenuated total reflection Fourier transform infrared (ATR-FTIR) and wide-angle X-ray diffraction (WAXD) analyses were performed. Figure 6 shows the ATR-FTIR spectra of silk/NR composites in the full range (a) and in the carbonyl region (b). The Silk_10_/NR_90_ composite (blue line) exhibited an amide I absorption peak centered at 1650 cm^–1^, indicating that silk adopted a random-coil conformation in the NR matrix [54]. Meanwhile, the formation of a β-sheet structure was confirmed for silk_20_/NR_80_ (pink line), as evidenced by the appearance of a shoulder peak at 1625 cm^–1^. The formation of the β-sheet structure was more enhanced in the case of silk_40_/NR_60_ (green line), and the relative intensity of the peak at 1625 cm^–1^ was larger than that of 1650 cm^–1^. The same trend was confirmed by WAXD measurements. The one-dimensional (1D) WAXD profiles of pure NR and silk_10_/NR_90_ had only the scattering of an amorphous halo, whereas the appearance of new peaks at 0.42, 0.36, and 0.22 nm, which corresponded to an antiparallel β-sheet structure, were confirmed for silk_20_/NR_80_ and silk_40_/NR_60_ (Figure 7, arrows); the Miller indices of these peaks were assigned as (020), (021), and (040), respectively [55,56,57]. Together with the results of the tensile tests and rheological measurements, the increase in modulus at higher silk content was attributed to not only the simple increase in silk volume but also the secondary structure change from random to β-sheet structures. As the highest mechanical performance was measured for silk_20_/NR_80_, the excessive formation of a β-sheet structure might disturb the enhancement of toughness in the silk/NR composite system, probably because the compatibility of these two components was lowered by the aggregation of silk.

### 2.3. Preparation of the DOPA-silk/NR Composite

Next, the composite of DOPA-modified silk and NR was prepared. The enzymatic DOPA modification of silk solution was conducted by following a previously developed method, as shown in Scheme 1 [30,58]. This DOPA-modified silk was further mixed with NR latex, to give DOPA-silk_20_/NR_80_ (Figure 8). The DOPA content in the silk solution was determined by amino acid analysis (Appendix A). The appropriate evaluation of DOPA content is quite important because it should affect the adhesion properties and the secondary structures of silk. Namely, the mechanical properties of DOPA-silk-based composites would be changed depending on it. It was calculated that 19% of tyrosine residue was converted to the DOPA structure, indicating that 0.96% of DOPA existed in the sequence. Even though this percentage is not very high, this small portion of DOPA has the potential to enhance the adhesion property of silk dramatically [30]. The silk content was fixed at 20 wt% because the largest fracture energy was observed with this content in the unmodified silk/NR composites. To clarify the effect of the DOPA structure on the composite material, the control sample without any tyrosinase added, silk_20_/NR_80_-2, was also prepared. Note that the concentration of the silk solution for the preparation of DOPA-silk_20_/NR_80_ and silk_20_/NR_80_-2 was diluted to 40 g/L because gelation took place during DOPA modification with the use of 80 g/L silk solution, whose concentration was used for the preparation of silk_20_/NR_80_. At high silk concentrations, the dityrosine structure was probably formed to some degree via the overoxidation of tyrosine residue to give chemically crosslinked silk-gel [59]. The film thicknesses of DOPA-silk_20_/NR_80_ and silk_20_/NR_80_-2 were approximately 0.20 mm, which was thinner than that of silk_20_/NR_80_ because of the difference in the silk concentration used.

### 2.4. The Mechanical Properties of DOPA-Silk/NR

Tensile tests of DOPA-silk_20_/NR_80_ and silk_20_/NR_80_-2 were performed, and their stress-strain curves are shown in Figure 9. As a result, there was no significant difference in the Young’s modulus, while both breaking strength and elongation improved after DOPA modification (Figure 10). In addition, the fracture energy of DOPA-silk_20_/NR_80_ was five times larger than that of silk_20_/NR_80_-2. These results definitively indicate the positive effect of the DOPA structure in fabricating silk composites with NR. We also found that the mechanical properties of silk_20_/NR_80_-2 were inferior to those of silk_20_/NR_80_, despite having the same silk/NR ratio. The difference in the starting silk concentration might affect several parameters, such as the miscibility of silk and NR, the stability of the latex particles, and the drying speed of the samples, resulting in the mechanical property difference of the formed composites. Meanwhile, the remaining of buffer salts and tyrosinase might also affect their mechanical properties. However, the effect of tyrosinase should be neglectable, because it did not improve the interfacial adhesion properties between substrates, according to the reference of Park and coworkers [60]. They mentioned that the adhesion properties of chitosan did not change in the presence or absence of tyrosinase, and thus it did not affect the mechanical properties of the final product, the silk/chitosan composite. Although the chemical structures of chitosan and NR are largely different, the interaction of less polar NR and tyrosinase should be smaller than that of chitosan and tyrosinase. The buffer salts might have influence on the mechanical properties; nevertheless, the use of buffer solution was essential for DOPA modification. These salts would be possible to remove by dialysis; however, we wanted to use DOPA-modified silk solution soon after the modification reaction finished because a report mentioned the spontaneous autoxidation of DOPA solution in the absence of any catalyst [61]. Therefore, we used DOPA-modified silk solution, without any purification, in this study. Although silk_20_/NR_80_-2 should be appropriate as a control for DOPA-silk_20_/NR_80_, the mechanical properties of silk_20_/NR_80_ and DOPA-silk_20_/NR_80_ were also compared to confirm that there was no significant difference between them (Appendix A). On the other hand, the stress–strain curves of DOPA-silk_20_/NR_80_ showed more elastic shapes compared to those of silk_20_/NR_80_ after the linear region, probably owing to the secondary structure changes of DOPA-modified silk, as mentioned below. In short, the introduction of the DOPA structure affected the mechanical properties of the formed composites, as well as changing the silk contents and the concentration of the silk solution. To evaluate the effect of DOPA more precisely, we further examined to prepare DOPA-silk_20_/NR_80_ with different DOPA content by reducing the tyrosinase amount from 1300 to 300 U/mL. After the tyrosinase treatment, amino acid analysis of resultant silk solution was performed to find that the conversion ratio did not change so largely, as we expected, despite the difference of enzyme load (Appendix A). The conversion ratio and DOPA content were determined as 24.5% and 1.08% at this condition, respectively. The tuning of DOPA contents would be possible by changing pH values; however, we need to be careful about the coagulation of latex particles [59], as well as the secondary structure change of silk [62]. As we found the certain difficulty to deal with this problem, we decided to report the further progress about the optimization of DOPA-modification reaction and successive fabrication of DOPA-silk-based composites with rubber in elsewhere. Rheological measurements were also conducted for silk_20_/NR_80_-2 and DOPA-silk_20_/NR_80_ (Figure 11 and Appendix A). The strain sweep measurements revealed that the linear viscoelastic region was observed within almost the same range, while the frequency sweep measurements gave coincident curves. These results agreed with the results of the tensile tests. Figure 12 shows cross-sectional SEM images of the composites. A relatively smoother surface was imaged for DOPA-silk_20_/NR_80_ compared to that of silk_20_/NR_80_-2 before stretching, suggesting that the compatibility of silk with NR was enhanced by DOPA modification.

### 2.5. Structural Characterization of the DOPA-Silk/NR Composite

The secondary structures of silk_20_/NR_80_-2 and DOPA-silk_20_/NR_80_ were characterized by ATR-FTIR and WAXD measurements. As the tyrosine residue in silk from *B. mori* mainly exists at the ends of the repetitive oligopeptide units (GAGAGY) that form a β-sheet structure, the modification of this residue would largely affect the formed structures. In fact, the slight expansion of *d*-spacing of β-sheet crystal domains was previously confirmed in our group by introducing polyphenylene ether chains at the tyrosine residues [24], while Zhao et al. reported the structure change from random to β-sheets via the conversion of tyrosine to azobenzene units [63]. Moreover, Kaplan and coworkers reported that the decoration of silkworm silk with silica-binding peptides and SiO_2_ by scaffolding the tyrosine units led to an increased amount of α-helical and random coil structures [64]. These results suggested that the secondary structure could be changed by various factors, including the conditions during the modification reaction (i.e., solvent, temperature, and pH), the bulkiness, and hydrophobicity/hydrophilicity of introduced units, and so on. In the current study, we performed the ATR-FTIR measurements of silk_20_/NR_80_-2 and DOPA-silk_20_/NR_80_ samples (Figure 13), and resulting bimodal peaks centered at 1625 and 1650 cm^–1^ were observed for silk_20_/NR_80_-2, and a unimodal peak at 1650 cm^–1^ was detected for DOPA-silk_20_/NR_80_.

Additionally, WAXD measurements revealed that silk_20_/NR_80_-2 showed peaks attributed to the β-sheet structure, although these peaks were absent for DOPA-silk_20_/NR_80_-2 (Figure 14a). These results clearly indicate that a random conformation was mainly adopted for DOPA-silk_20_/NR_80_, whereas the β-sheet structure was formed in silk_20_/NR_80_-2, despite having the same silk content and preparation process. This secondary structure change should contribute to the enhancement of the mechanical properties of DOPA-silk_20_/NR_80_, as well as the higher adhesion properties of the DOPA structure. The stronger interaction between DOPA and the hydrophilic α-terminus of NR [65] might also help to improve the mechanical properties of the formed composite, although no clear evidence for this speculation has been obtained so far. After the stretching deformation, the WAXD 1D profiles of silk_20_/NR_80_-2 and DOPA-silk_20_/NR_80_ demonstrated the clear strain-induced crystallization of NR, according to previous studies and assignment [66,67]. This result indicates that the silk/NR composite samples studied here can show the strain-induced crystallization, which is essential to achieve the NR-specific physical properties. In other words, silk molecules blended with NR do not disturb the NR’s original crystallization. Overall, DOPA modification positively affected the fabrication of the silk/NR composite.

## 3. Materials and Methods

### 3.1. Materials

Tyrosinase from mushroom (≥1000 U/mg solid) was purchased from Sigma-Aldrich (St. Louis, MO, USA). NR latex with low ammonia content (product name: Ulacol^®^) was purchased from HR-TEC (Osaka, Japan). The composition of NR latex used in this study was as follows: NR, 61.5 wt%; ammonia, 0.1 wt%; and H_2_O, 38.4 wt%. The other chemicals were used as received, without purification, unless otherwise noted.

### 3.2. Preparation of Silk Solution

A silk solution from silkworm cocoons of *B. mori* was prepared by a previously reported protocol [5]. Briefly, silkworm cocoons were boiled for 20 min in a 0.02 M Na_2_CO_3_ solution and subsequently washed with Milli-Q water. After drying for 24 h at 25 °C, extracted residues were dissolved in a 9.3 M LiBr solution at 60 °C for 2 h. The solution was further dialyzed with Milli-Q water for 72 h, using a dialysis membrane (Pierce Snake Skin MWCO 3500; Thermo Fisher Scientific, Waltham, MA, USA). The concentration of silk solution was determined by drying 1.0 mL of solution and weighing the resultant mass.

### 3.3. Preparation Protocol of Silk/NR Composites

To the silk solution (80 g/L, 1.0 mL), NR latex solution (1.3 mL) was added and stirred at 25 °C for 30 min. The mixture was poured into a dumbbell-shaped silicon mold of the Japanese Industrial Standard (JIS K 6251) and dried in the fume hood for 24 h, to give silk/NR composites with 10 wt% silk content (Silk_10_/NR_90_). The composites with other silk contents were prepared by changing the volume of silk solution.

### 3.4. Preparation Protocol of DOPA-Silk/NR Composites

Enzymatic modification of tyrosine residue in silk with DOPA was conducted by a previously reported protocol with a slight modification [30,58]. Briefly, to a silk solution (40 g/L, 4.0 mL), 0.1 M phosphate buffer solution (4.0 mL, pH 7.0) and tyrosinase (final concentration: 1300 U/mL) were added and stirred at 25 °C for 1 h in open-air conditions. This solution was directly mixed with NR latex solution (1.3 mL), without any purification, and stirred at 25 °C for 30 min. The mixture was poured into a dumbbell-shaped silicon mold of JIS K 6251 and dried in the fume hood for 24 h, to give a DOPA-silk/NR composite with 20 wt% silk content (DOPA-Silk_20_/NR_80_). The control sample was also prepared without the use of tyrosinase (Silk_20_/NR_80_-2). The DOPA content was evaluated by amino acid composition analysis, namely, the ninhydrin method [68]. The hydrolyzed amino acids were characterized by using high-speed amino acid analyzers, L-8900 and L8500A (Hitachi-HighTech, Tokyo, Japan). In addition to the natural amino acids, DOPA (Sigma-Aldrich) was used to calibrate the analyzer, and, accordingly, the peak was assigned as DOPA (see Appendix A).

### 3.5. Tensile Tests

The mechanical properties of the composites were evaluated by a tensile testing machine (EZ-Test, Shimadzu, Kyoto, Japan) with an elongation rate of 10 mm/min. The measurements were performed at approximately 40% relative humidity and 25 °C. The Young’s modulus, breaking strength, elongation at break, and fracture energy were analyzed based on the stress–strain curves of the formed composites. The experiments were conducted at least 5 times for every condition.

### 3.6. Rheological Measurements

The samples were set between the parallel plate of a rheometer (MCR 102 Anton Paar, Austria). *G’* and *G’’* were measured at 25 °C. The strain was fixed at 0.1% for frequency dependency measurements, whereas the frequency was fixed at 10 Hz for sweep dependency measurements.

### 3.7. SEM Observations

The morphology of the formed composites was analyzed by SEM. The samples were mounted onto an aluminum stub, sputter-coated with gold, and imaged by SEM (JSM6330F, JEOL Ltd., Tokyo, Japan) at an accelerating voltage of 5 kV. The cross-sectional surface of the as-prepared samples was prepared by using a razor blade.

### 3.8. ATR-FTIR Measurements

ATR-FTIR spectra were measured on an IRPrestigae-21 Fourier transform infrared spectrophotometer (Shimadzu Corporation, Kyoto, Japan) with a MIRacle A single reflection ATR unit using a Ge prism. The measurements were conducted from 4000 to 800 cm^–1^. The background spectra obtained under the same conditions were subtracted from the scan for each sample.

### 3.9. WAXD Measurements

The synchrotron WAXD measurements were performed on the BL40XU and BL05XU beamlines of SPring-8, Harima, Japan, according to a previous study [69]. The X-ray energy was 12.4 keV at a wavelength of 0.1 nm, the sample-to-detector distance for the WAXD measurements was approximately 275 mm, and the exposure time for each diffraction pattern was 1 s. The measurement was performed three times for each sample, and the resultant three two-dimensional WAXS images were composited. The obtained two-dimensional (2D) scattering patterns were converted into 1D profiles by using Fit2D software (European Synchrotron Radiation Facility (ESRF), Grenoble, France) [70]. Simultaneous WAXD measurements during tensile deformation were performed by using the film samples placed on a load cell (20 N) by clamping at both edges, according to previous studies [57,69]. The stretching rate was 2 mm min^−1^. The synchrotron WAXD measurement was initiated at the threshold of the stress-strain curve and repeated at 5-second intervals, until the films fractured.

### 3.10. Statistical Analysis

The significant differences in the studies of the mechanical properties were determined by (un)paired *t*-tests with a two-tailed distribution. Differences were considered statistically significant at *p* < 0.01 (*).

## 4. Conclusions

Silk and NR composites were successfully prepared by mixing degummed silk solution from *B. mori* silkworm and NR latex. The significant enhancement of mechanical properties was confirmed for silk/NR composites compared to a NR-only product. In particular, good elastic silk-based materials could be fabricated by combining silk and NR, which have largely different characteristics. The largest fracture energy of 35 MJ/m^3^ was obtained by adding 20 wt% silk solution. The Young’s modulus and elongation were 38 MPa and 750% at this condition, respectively. Additionally, DOPA modification of the tyrosine residue of silk improved the toughness of the formed composite. The fracture energy of DOPA-silk_20_/NR_80_ was 4.6 times larger than that of control. The efficient adhesion property of DOPA residue, together with the secondary structure change, was believed to be important for this improvement. The knowledge obtained herein should contribute to the development of the fabrication of novel silk-based elastic materials.

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
