# Peer review of "Silk/Natural Rubber (NR) and 3,4-Dihydroxyphenylalanine (DOPA)-Modified Silk/NR Composites: Synthesis, Secondary Structure, and Mechanical Properties"

_molecules, 2020, doi:10.3390/molecules25010235_

Round 1
Reviewer 1 Report
the introduction section was hard to understand the novelty of the work and the privilege of this study in comparison with similar work. please clear the importance of this work only and the gap between the similar work dealing with silk/NR composite.
please consider changing line 50 for latex gloves
According to lines 72-80, the authors indicated that there is some similar work reporting the mechanical enhancement of the silk-NR composite however they wanted to ''the mechanical properties of the formed composites were not adequately discussed at the molecular level'', their aim is not clear. please consider rewriting this sentence.
line 117, ''The DOPA content was evaluated by amino acid composition analysis, namely, the ninhydrin method'', what is the importance of this analysis?
for the tensile test, the size of the samples and preferably the standard used for this test are better to be mentioned. please read the following article for this purpose: https://www.sciencedirect.com/science/article/pii/S1751616118301541?via%3Dihub
please improve the quality of figures 6, 7 and 13
please consider reporting some numerical result in the conclusion section
Reviewer 2 Report
In this report, the authors prepared a DOPA-modified silk-rubber composites that offered better mechanical properties to those of the unmodified composite. The most serious issues are the bad quality of figures. In figure 6,7 and 13, many curves on the figures are missing, which makes it very hard to evaluate these data. Besides, figure 3, 7, 10 and 14, the words on x-axis are hard to read. The some of the axis for figure 2 and 9 are also missing. The authors should also carefully check all the figures thoroughout the paper to make sure they have the same font and format.
1. The author should also run statistical analysis for figure s3.
2 On the SEM image (Figure 5), what is the word "code" stand for on top of the images?
3. In the preparation DOPA-silk/NR composites, the solution was directly dried without any purification. Did the author consider the effects from the residue buffer salts and tyrosinase? These may also affect the mechanical property of the composites.
4. Since the hypothesis "strong interaction between DOPA and NR help to improve the mechanical properties" lacks clear evidence, can the author change the dopa content as a way of showing the postive role of DOPA.
In summary, I can't recommend it to be pulished in the current status, but I am happen to review it again once this big issue has been corrected.
Round 2
Reviewer 1 Report
The authors have addressed the major concerns and I think the manuscript is now publishable.
Reviewer 2 Report
The authors have made corrections. I would recommend it to be published now.